# Joint Spatio-Temporal-Frequency Representation Learning for Improved Sound Event Localization and Detection

**DOI:** 10.3390/s24186090

**Published:** 2024-09-20

**Authors:** Baoqing Chen, Mei Wang, Yu Gu

**Affiliations:** 1School of Information and Communication, Guilin University of Electronic Technology, Guilin 541004, China; bqchen@mails.guet.edu.cn (B.C.); yu.gu@mails.guet.edu.cn (Y.G.); 2College of Physics and Electronic Information Engineering, Guilin University of Technology, Guilin 541004, China

**Keywords:** sound event localization and detection, spatial audio, spatio-temporal-frequency fusion, time-frequency alignment, SimAM

## Abstract

Sound event localization and detection (SELD) is a crucial component of machine listening that aims to simultaneously identify and localize sound events in multichannel audio recordings. This task demands an integrated analysis of spatial, temporal, and frequency domains to accurately characterize sound events. The spatial domain pertains to the varying acoustic signals captured by multichannel microphones, which are essential for determining the location of sound sources. However, the majority of recent studies have focused on time-frequency correlations and spatio-temporal correlations separately, leading to inadequate performance in real-life scenarios. In this paper, we propose a novel SELD method that utilizes the newly developed Spatio-Temporal-Frequency Fusion Network (STFF-Net) to jointly learn comprehensive features across spatial, temporal, and frequency domains of sound events. The backbone of our STFF-Net is the Enhanced-3D (E3D) residual block, which combines 3D convolutions with a parameter-free attention mechanism to capture and refine the intricate correlations among these domains. Furthermore, our method incorporates the multi-ACCDOA format to effectively handle homogeneous overlaps between sound events. During the evaluation, we conduct extensive experiments on three de facto benchmark datasets, and our results demonstrate that the proposed SELD method significantly outperforms current state-of-the-art approaches.

## 1. Introduction

Sound event localization and detection (SELD) has played a key role in advancing technologies for environmental monitoring [1], healthcare [2], bioacoustic monitoring [3], multimedia content search and retrieval [4], and autonomous driving [5]. In particular, sound events usually occur in unpredictable and unstructured contexts, characterized by ambient noise, reverberation, and interference. Moreover, such events are frequently non-stationary and exhibit high intra-class variability. These complexities present significant challenges for those interested in utilizing SELD in real-life scenarios.

To address these challenges, many studies have explored 1D and 2D convolution-based approaches for the detection and localization of sound events in spatial audio recordings. For example, a 1D convolutional neural network is constructed within an end-to-end training framework to directly recognize the classes and locations of sound events from raw waveform files [6]. The 2D convolution-based methods are effective at extracting time-frequency patterns within each feature map and learning certain inter-map patterns from the combined features after pre-processing operations [7,8,9], making them more powerful than methods based on 1D convolutional neural networks. The effectiveness of these methods originates from maintaining proper time-frequency (TF) alignment between feature maps, which are created using different frequency distributions from various sound sources over time. This alignment ensures that the network learns meaningful patterns across frequencies, which can help accurately associate target classes with the location of multiple concurrent sound events.

Although these 1D or 2D convolution methods have seen some progress, none of them are fully compatible with the spatio-temporal-frequency volumes intrinsic to multichannel audio data. This incompatibility makes it difficult to explicitly model the interrelationships among audio channels, limiting the effective use of spatial information and resulting in incomplete sound event representations. The issue is exacerbated in cases of concurrent sound events, where spatial locations may be swapped or misinterpreted, thereby compromising the overall performance of SELD. Furthermore, a “reality gap” often emerges when algorithms trained on synthetic spatial audio datasets, such as the TAU NIGENS Spatial Sound Event (TNSSE) 2021 dataset [10], are applied to real-world audio data. These algorithms typically experience a decline in performance due to the acoustic differences and variability present in real-world environments, which differ markedly from those in synthetic datasets.

There has also been a lot of interest in enhancing convolutional neural networks with forms of an attention module for the SELD task [9,11,12,13]. These modules complement backbone or head networks by encoding categorical and directional information of sound events. In practice, the refinement of sound event features is frequently conducted along the feature channel [14] or the time-frequency dimension [15] and then progressively through a cascade of both dimensions [16,17]. When multidimensional dependencies are not considered concurrently, the model’s ability to extract more discriminative information from volumetric data may be constrained. Apart from that, most attention modules are parametric, introducing extra parameters that heighten the risk of overfitting, particularly in small or highly variable datasets.

In response to these concerns, this paper introduces a novel framework for polyphonic SELD that utilizes the newly developed Spatio-Temporal-Frequency Fusion Network (STFF-Net) to jointly learn comprehensive features of sound events across spatial, temporal, and frequency domains. Figure 1 presents an overview of the proposed SELD framework. We start by constructing the input volume for MIC and FOA formats from the recorded multichannel audio signals. The feature volume, after augmentation, is then fed into our STFF-Net, which uses a sequence of Enhanced-3D (E3D) residual blocks to extract and refine spatio-temporal-frequency features of sound events through the integration of 3D convolution with a non-parametric attention mechanism. Furthermore, we leverage the multi-ACCDOA format to enhance the network’s ability to handle overlapping sounds from the same class, making it more robust in diverse acoustic scenarios. During the evaluation, we conduct extensive experiments on both synthetic and real spatial audio datasets to assess the performance of our proposed framework. The experimental results demonstrate that our framework outperforms current state-of-the-art (SOTA) works by a large margin.

In a nutshell, the main contributions of this paper are as follows:We, for the first time, conceptualize sound events as spatio-temporal-frequency entities, similar to actions in videos. This charts a new research trajectory for the SELD task.We propose a novel STFF-Net architecture that cohesively integrates the detection and localization of sound events, simplifies the polyphonic SELD optimization process, and provides a spatio-temporal analysis of sound events.We conduct comprehensive experiments on three public synthetic and real spatial audio datasets to validate the effectiveness of our proposed method.

The remainder of this paper is organized as follows. Section 2 reviews existing works, particularly those related to our approach. Section 3 introduces the details of our newly developed STFF-Net for polyphonic SELD. Section 4 describes the experimental setup and presents the results. Finally, the conclusions related to this work are given in Section 5.

## 2. Background and Related Works

### 2.1. Sound Event Localization and Detection

In the past decade, deep neural networks (DNNs) have emerged as the leading technique for sound event detection (SED) and sound source localization (SSL) [18,19]. Researchers have developed DNN-based approaches to jointly model the detection and localization of sound events, known as SELD, to gain a more complete understanding of environmental sounds [20,21]. In particular, polyphonic SELD has garnered substantial research attention due to its close resemblance to real-life scenarios. For example, Adavanne et al. [20] introduced SELDnet, which uses a shared backbone network to perform SED and direction of arrival (DOA) estimation. Cao et al. [22] proposed a two-stage strategy that utilizes separate networks for SED and DOA estimation and filters the DOA output based on the SED results. Despite the success, the fundamental interaction between SED and DOA estimation is not fully exploited. To address this issue, Cao et al. [11] later designed Event-Independent Network V2 (EINV2) with soft-parameter sharing between SED and DOA estimation to improve the interaction and the track-wise output format to handle homogeneous overlapping events. Hu et al. [9] further enhanced the performance of their model by incorporating dense blocks and replacing multi-head self-attention with conformer. As another direction for improvement, Shimada et al. [23] proposed a unified format called Activity-Coupled Cartesian Direction of Arrival (ACCDOA) that fuses SED and DOA estimation into a 3D Cartesian regression task to perform SELD.

In addition, effective feature engineering has been studied recently for the SELD task. For example, Nguyen et al. [24] developed the Spatial Cue-Augmented Log-Spectrogram (SALSA) to enhance polyphonic SELD by accurately mapping signal power with source directional cues for MIC and FOA formats. They then proposed SALSA-Lite [25], which is a computationally optimized variant for the MIC format that employs normalized cross-channel phase differences to enhance efficiency and preserve performance. Rosero et al. [26] improved SELD feature extraction with bio-inspired Gammatone filters and utilized a temporal convolutional network to better capture long-term dependencies compared to previous works. Huang et al. [27] transformed audio signals into graph representations and developed a graph convolutional network to simultaneously learn the spatio-temporal features of sound events.

Distinct from these existing works, our spatio-temporal-frequency fusion network employs a novel model architecture that accurately characterizes sound events by jointly capturing comprehensive features across spatial, temporal, and frequency domains in multichannel audio recordings. Consequently, our model significantly outperforms SOTA methods on several benchmark datasets.

### 2.2. Attention Mechanisms

The attention mechanism can be viewed as a dynamic selection process that adaptively weights features according to their importance. This approach has proven beneficial in numerous computer vision tasks [28]. One successful example is the Squeeze-and-Excitation (SE) attention module, which establishes interdependencies between different feature channels [14]. The following works further enhance their performance by incorporating long-range dependencies and by extracting detailed contextual information through more advanced modules, such as the Global Context (GC) module and the Convolutional Block Attention Module (CBAM) [16,17]. However, these methods generate 1D or 2D weights that uniformly treat neurons across spatial or channel dimensions of the image features, as shown in Figure 2a,b. This uniformity restricts their ability to extract more discriminative information from volumetric data that contain multiple dimensions.

In subsequent research endeavors, multidimensional dependencies are simultaneously captured to facilitate feature learning. For example, the residual attention network [29] learns 3D weights by integrating sub-networks into its different layers. However, the network’s scalability is limited by heightened computational complexity and resource demands. In contrast, SimAM [30], as depicted in Figure 2c, is a simple, parameter-free attention mechanism that directly generates 3D weights without requiring extensive structural tuning of the original networks. These 3D attention maps align with the temporal, frequency, and spatial dimensions of sound sources, enabling SimAM to enhance the 3D feature volume of sound events effectively while maintaining a low computational cost. Motivated by these existing works, our approach leverages SimAM to refine the spatio-temporal-frequency features extracted by the 3D convolutional network, allowing us to acquire precise and detailed characterizations of sound events recorded in multichannel audio signals.

## 3. Methodology

In this section, we first introduce our newly developed SELD network and then present its related components, such as the spatio-temporal-frequency volume, the Enhanced-3D residual block, and the class- and track-wise output format.

### 3.1. Spatio-Temporal-Frequency Fusion Network

The spatial, temporal, and frequency properties of sound sources are usually interdependent, necessitating a comprehensive analysis across these domains to accurately characterize sound events in multichannel audio recordings. This analysis enables the establishment of correlations between sound event classes and their corresponding locations, which is crucial for resolving overlapping events. To this end, we propose a novel STFF-Net model to improve the performance of polyphonic SELD, as illustrated in Figure 3. This model integrates detection and localization of sound events within a unified framework, which consists of a customized backbone network, basic bi-directional gated recurrent units (BiGRUs), and a multi-ACCDOA output format.

Specifically, the STFF-Net model begins by receiving a spatio-temporal-frequency volume with dimension C×D×T×F, where *D* is the depth defined by the number of feature channels, *T* is the number of time frames, and *F* is the number of frequency bins. Because each audio feature map is represented as a grayscale image with a single-channel intensity, we initialize the channel dimension *C* to 1. The volume is then passed through two 3D convolution layers and a 3D average pooling layer to decrease the resolution of the feature maps while preserving their key characteristics. Next, we deploy a sequence of Enhanced-3D residual blocks in the network to extract specialized representations, as shown on the right side of Figure 3. In particular, each Enhanced-3D residual block consists of two 3D convolutional layers enhanced with SimAM, allowing the network to focus on significant sound events. We set the stride size of the first convolutional layer to 1 or 2. Skip connections are utilized for convolutional layers to ensure that the gradient can flow directly within the entire network. This design helps the model converge fast while simultaneously reducing the probability of overfitting during the training process [31]. Later, we pool the extracted high-level features along the frequency axis, but the other dimensions are kept fixed. To capture temporal context information in both forward and backward directions, we reshape the outputs and then feed them into two BiGRU layers with 256 units [32]. Our network ends with fully connected (FC) layers, in which the first layer includes 256 units and the second layer includes $n_{\mathrm{tracks}}\times n_{\mathrm{classes}}\times 3$. After further upsampling to match the target annotation’s resolution, we obtain the final shape of $\left(T/6,n_{\mathrm{tracks}}\times n_{\mathrm{classes}}\times 3\right)$, which indicates a 6× reduction of the temporal dimension and provides ACCDOA vector coordinates for each target class per track.

### 3.2. Spatio-Temporal-Frequency Volume

The spatio-temporal-frequency volume is a 3D representation that encompasses the spatial, temporal, and frequency properties of sound sources in multichannel audio signals. This 3D tensor consists of depth, height, and width dimensions, where depth represents spatial characteristics determined by differences between audio channels, height reflects the evolution of sound sources over time, and width represents the spectral content of the audio signal. Typically, SED relies on time-frequency patterns to differentiate sound classes, while SSL uses amplitude and/or phase differences between microphones to estimate spatial locations. To carry out SELD tasks, we construct a volume that combines SED and SSL features. In detail, we extract multichannel log-linear spectrograms and frequency-normalized inter-channel phase differences (NIPDs) [25] for the MIC format and linear-scale intensity vectors (IVs) [33] for the FOA format. These features for each format are then sequentially stacked in an audio channel order to form a 3D feature volume aligned in all three dimensions, as illustrated in Figure 4a,b. The resulting volumes, referred to as SALSA-Lite for MIC and LinSpecIV for FOA, serve as initial inputs for our STFF-Net model.

Mathematically, the log-linear spectrogram of a spatial audio recording with *M* channels is defined as follows:(1)$$\mathrm{LinSpec}\left(t,f\right)=\log\left({\left|X_{1:M}\left(t,f\right)\right|}^{2}\right)\in R^{M\times T\times F},$$
where $X_{i}\left(t,f\right)$ is the short-time Fourier transform (STFT) of the audio signal from the *i*-th channel, *t* is the time frame index, *f* is the frequency index, and $\left|\;\cdot \;\right|$ denotes the element-wise complex modulus. To offer directional cues to the sound source, we compute the NIPD in SALSA-Lite as follows:(2)$$\Lambda \left(t,f\right)=-v{\left(2\pi f\right)}^{-1}\arg\left[X_{1}^{*}X_{2:M}\left(t,f\right)\right]\in R^{M-1},$$
where *v* is the sound velocity, and ${\left(\;\cdot \;\right)}^{*}$ is the conjugate transpose. Similarly, IV in LinSpecIV is obtained by:(3)$$\begin{array}{c} I\left(t,f\right)=\left[\begin{array}{c} I_{X}\left(t,f\right)\\ I_{Y}\left(t,f\right)\\ I_{Z}\left(t,f\right) \end{array}\right]=\frac{1}{\rho_{0}v}ℜ\left\{X_{W}^{*}\left(t,f\right)\left[\begin{array}{c} X_{X}\left(t,f\right)\\ X_{Y}\left(t,f\right)\\ X_{Z}\left(t,f\right) \end{array}\right]\right\},\\ I_{\mathrm{norm}}=-\frac{I(t,f)}{{∥I\left(t,f\right)∥}_{2}}\in R^{3}, \end{array}$$
where $\rho_{0}$ is the sound density, W is the omnidirectional component of the FOA format, and X, Y, and Z correspond to its directional components in the front–back, left–right, and up–down axes, respectively [33,34]. Additionally, $ℜ\left\{\;\cdot \;\right\}$ is the real part, and ${∥\;\cdot \;∥}_{2}$ is the $\ell_{2}$ norm of a vector. Table 1 outlines the SELD features used in this study to assess the performance of our chosen initial feature volumes.

### 3.3. Enhanced-3D Residual Block

Given that standard 1D or 2D convolution methods are unable to adequately capture the features of sound events across spatial, temporal, and frequency domains, we developed a new building module called the Enhanced-3D (E3D) residual block to replace 2D residual units within the residual learning framework. Formally, we define an E3D residual block as follows:(4)$$y=\mathrm{Concat}\left(\left[E_{1}⊙F_{1}\left(x\right),E_{2}⊙F_{2}\left(x\right),\ldots ,E_{C}⊙F_{C}\left(x\right)\right]\right)+H\left(x\right).$$
Here, x and y are the input and output tensors of the building block, respectively; $F_{c}\left(x\right)$ is the portion of the feature volumes generated after the 3D convolutional layers corresponding to the *c*-th channel; $E_{c}$ indicates the 3D attention weights for the feature volume $F_{c}\left(x\right)$; ⊙ denotes element-wise multiplication; $\mathrm{Concat}\left(\;\cdot \;\right)$ denotes the concatenation operation along the feature channel dimension; H(x) is the shortcut connection that preserves the input information; and H(x)=x is an identity mapping.

The E3D residual block capitalizes on two key technologies. The first is the encoding of spatio-temporal-frequency information of sound events through 3D convolution and 3D pooling operations. 3D convolution [35] jointly captures event-level patterns across spatial, temporal, and frequency domains by convolving a 3D kernel with the feature volumes. Formally, the value of a unit at position (d,t,f) in the *j*-th feature volume in the *l*-th layer, denoted as $z_{lj}^{dtf}$, is given by:(5)$$z_{lj}^{dtf}=\mathrm{ReLU}\left(\sum_{m}\sum_{\alpha =0}^{A-1}\sum_{\beta =0}^{B-1}\sum_{\gamma =0}^{\Gamma -1}w_{ljm}^{pqr}z_{(l-1)m}^{\left(d+\alpha )(t+\beta )(f+\gamma \right)}+b_{lj}\right),$$
where A, B, and Γ are the sizes of the kernel along the depth, height, and width dimension, $b_{lj}$ is the bias for the given feature volume, *m* indexes over the set of feature volumes in the (l−1)-th layer connected to the current feature volume, and $w_{ljm}^{pqr}$ is the value at position (p,q,r) of the kernel connected to the *m*-th feature volume in the preceding layer. As the kernel slides across the entire input volume, a complete 3D feature volume is produced.

The second key technology is the integration of a non-parametric attention module using SimAM [30] to generate 3D attention weights, which emphasize the informative and discriminative features of sound events simultaneously in three dimensions. These attention weights for feature volumes are determined based on the energy of the corresponding neuron. The energy function for each neuron $n_{o}$ is given by:(6)$$e_{o}^{★}=\frac{4\left(\sigma^{2}+\lambda \right)}{{\left(n_{o}-\mu \right)}^{2}+2\sigma^{2}+2\lambda },$$
where $\mu =\frac{1}{\Delta }\sum_{o=0}^{\Delta -1}n_{o}$ and $\sigma^{2}=\frac{1}{\Delta }\sum_{o=0}^{\Delta -1}{\left(n_{o}-\mu \right)}^{2}$ are the mean and variance at a certain depth in a feature volume, Δ=T×F is the number of neurons at that depth, and λ is a regularization parameter. In fact, a neuron $n_{o}$ with lower energy $e_{o}^{★}$ is more distinct from the surrounding neurons, which makes it more crucial for feature processing. Therefore, the importance of each neuron is quantified as $1/e_{o}^{★}$. For each feature volume, we can obtain 3D attention weights by aggregating $1/e_{o}^{★}$ over depth, height, and width dimensions and then applying a sigmoid function to constrain overly large values without changing the relative relevance of individual neurons. The attention module is straightforward and intuitive, with all computations being element-wise except for the mean μ and variance $\sigma^{2}$ calculations along the depth dimension of the feature volumes, as shown in Algorithm 1. In essence, the whole refinement phase is analogous to beamforming for each feature volume, but it does not increase network parameters.
**Algorithm 1** PyTorch-like Implementation of a Non-parametric Attention Module**Input**:
a batch of extracted feature volumes $Z\in R^{B\times C\times D\times T\times F}$, coefficient λ as defined in Equation (Equation 6)**Output**:
the refined feature volumes $\hat{Z}\in R^{B\times C\times D\times T\times F}$1:**def** forward (Z, lambda):2:    # time-frequency size3:    Δ = Z.shape[3] ∗ Z.shape[4] − 14:    # square of (n − u)5:    d = (Z − Z.mean(dim=[3, 4])).pow(2)6:    # d.sum() / n is depth variance of feature volumes7:    v = d.sum(dim=[3, 4]) /Δ8:    # E_inv groups all importance of feature volumes9:    E_inv = d / (4 ∗ (v + lambda)) + 0.510:  # return the attention-weighted feature volumes11:  **Return** Z ∗ sigmoid(E_inv)

### 3.4. Class- and Track-Wise Output Format

We adopt the multi-ACCDOA format for the model outputs, which merges the activity and DOA of each sound event into a single vector and solves cases with homogeneous overlap in polyphonic SELD. In addition, class-wise auxiliary duplicating permutation-invariant training (ADPIT) is utilized to provide consistent training targets and avoid interference from zero vectors [36]. Overall, the loss function of our STFF-Net model can be defined as follows:(7)$$\begin{array}{c} L^{\mathrm{PIT}}=\frac{1}{ST}\sum_{s}^{S}\sum_{t}^{T}\min_{\alpha \in \mathrm{Perm}(st)}l_{\alpha }^{\mathrm{ACCDOA}},\\ l_{\alpha }^{\mathrm{ACCDOA}}=\frac{1}{N}\sum_{n}^{N}\mathrm{MSE}\left(P_{\alpha ,n}^{*},{\hat{P}}_{n}\right), \end{array}$$
where *N* and *S* denote the number of tracks and sound classes, respectively. Given class *s* and frame *t*, α∈Perm(st) denotes one possible class-wise frame-level permutation, $P_{\alpha ,n}^{*}$ denotes an ACCDOA target of a permutation α at track *n*, and ${\hat{P}}_{n}$ denotes the corresponding ACCDOA prediction. We set the parameter *N* to 3, as it is fairly common for up to three events to occur concurrently. During inference, if the magnitude of event vectors exceeds the preset threshold, we view them as active events and then choose the Cartesian DOAs accordingly.

## 4. Evaluation

In this section, we conduct extensive experiments on both synthetic and real spatial audio datasets to validate the effectiveness of our newly developed STFF-Net. For fair comparisons, we modify the STFF-Net by replacing the 3D convolution and pooling layers with 2D counterparts and by removing the attention module, creating a simpler baseline called CRNN. We also use consistent experimental settings to rigorously test performance.

### 4.1. Experimental Setup

#### 4.1.1. Datasets

The models in this work are trained and evaluated using three benchmark datasets: TNSSE 2021 [10], STARSS 2022 [37], and L3DAS22 Task 2 [38]. Table 2 summarizes the main characteristics of these datasets.

TNSSE 2021 dataset: This synthetic spatial audio dataset was released for Task 3 of the DCASE 2021 Challenge. It includes a public set of 400, 100, and 100 one-minute audio recordings for training, validation, and testing, respectively, as well as a private set of 200 one-minute recordings for evaluation. These recordings are crafted by convolving dry audio signals of sound events with multichannel room impulse responses (RIRs) with DOA labels and then adding spatial ambient noise. To replicate real-world conditions, the RIR and noise samples are collected from 15 distinct interior locations, with a balanced distribution of static and moving sound events across the range of azimuth $\left[-180^{\circ },180^{\circ }\right)$ and elevation $\left[-45^{\circ },45^{\circ }\right]$. Additional directional interference from static or moving non-target sound sources is also included in the dataset. Due to its accessibility, well-defined partitions, and relative complexity, this dataset is used in the majority of our experiments.

STARSS 2022 dataset: This dataset marks a significant advancement over the above dataset by utilizing real-world audio recordings with spatiotemporal labels, which are acquired in real time from motion trackers. The dataset consists of audio recordings ranging from 30s to 6 min, divided into sections for development and evaluation. The development set comprises 67 training clips and 54 testing clips, with a total duration of approximately 4 h 52 min. Audio data collection was conducted in 11 different rooms located in Tokyo and Tampere, where sound events were recorded within azimuth and elevation ranges of $\left[-180^{\circ },180^{\circ }\right)$ and $\left[-90^{\circ },90^{\circ }\right]$, respectively. The diversity and density of these events vary greatly, providing a wide range of natural sound distributions. Despite the presence of four or five overlapping events, they only account for 1.8% of the entire dataset.

L3DAS22 Task 2 dataset: This synthetic dataset was provided in the L3DAS22 Challenge for 3D SELD. Unlike the above challenges, this one requires the estimation of exact locations of sound events. The dataset comprises 7.5h of multiple-source and multiple-perspective (MSMP) Ambisonics recordings, each of which lasts 30s. These recordings are divided into 600 recordings for training, 150 recordings for validation, and 150 recordings for testing. Each subset includes an equal number of files that correspond to one, two, and three overlapping sound events. The RIRs are collected from 252 fixed spatial points in an office room that measures about 6 meters in length, 5 m in width, and 3 m in height. Two FOA microphones are positioned centrally within the room.

#### 4.1.2. Data Augmentation

Due to the limited number of samples in both synthetic and real spatial audio datasets, we utilize three data augmentation strategies to enhance data diversity and improve model generalization, including frequency shifting [24], hybrid cutout [39,40], and spatial augmentation [41,42]. Note that only spatial augmentation alters the ground truth, while frequency shifting and hybrid cutout do not.

Frequency shifting is used to mitigate the impact of noise and simulate different microphone distances by changing the frequency content of the audio signal. To do this, we randomly shift the input feature volumes upwards or downwards along the frequency axis by up to 10 bands. Hybrid cutout consists of two operations for preventing overfitting, namely, random cutout and TF masking. Random cutout produces a rectangular mask on the input feature volumes, while TF masking produces a cross-shaped mask. Each operation has an equal probability of occurring, with only one being executed at a time. In addition, spatial augmentation is performed by swapping channels in the input feature volume for MIC and FOA formats, which support 16 and 8 audio channel swapping methods, respectively. To ensure consistency, target labels are adjusted accordingly. This technique increases the variety of spatial locations of sound events within the spatial audio dataset.

#### 4.1.3. Evaluation Metrics

To evaluate SELD performance, we adopt the official metrics introduced in the DCASE and L3DAS22 Challenges [21,43]. These metrics provide a joint measure of localization and detection, including location-dependent F1-score $\left(F_{\leq \tau }\right)$, error rate $\left(ER_{\leq \tau }\right)$, class-dependent localization recall $\left(LR_{\mathrm{CD}}\right)$, and localization error $\left(LE_{\mathrm{CD}}\right)$. Here, τ denotes the spatial threshold, which is set at $20^{\circ }$ for DCASE and 2 m for L3DAS22, respectively, while $F_{\leq \tau }$ and $ER_{\leq \tau }$ consider true positive predictions that are less than a spatial threshold τ from the ground truth. Additionally, $LR_{\mathrm{CD}}$ and $LE_{\mathrm{CD}}$ only calculate localization predictions when the detected sound class is correct. We also compute an aggregated SELD error metric as a unified measure for the overall evaluation, defined as follows:(8)$$E_{\mathrm{SELD}}=\frac{1}{4}\left[ER_{\leq \tau }+\left(1-F_{\leq \tau }\right)+\frac{LE_{\mathrm{CD}}}{180^{\circ }}+\left(1-LR_{\mathrm{CD}}\right)\right],$$
where $E_{\mathrm{SELD}}$ is used to select the model and its hyperparameter. An optimal SELD system is characterized by low $ER_{\leq \tau }$, high $F_{\leq \tau }$, low $LE_{\mathrm{CD}}$, high $LR_{\mathrm{CD}}$, and a low overall $E_{\mathrm{SELD}}$ metric.

Moreover, we utilize both micro- and macro-averaged evaluation metrics, unlike most previous studies that have used either one or the other. This dual approach allows for a more comprehensive and balanced evaluation by considering both the influence of classes with larger sample sizes and the performance of smaller classes.

#### 4.1.4. Parameter Configuration

We follow a standard training pipeline for all models. Specifically, we use a sampling rate of 24 kHz, Hann window of 1024 samples, hop length of 400 samples, and 1024 FFT points, resulting in the input frame rate of 60 fps. The model temporally downsamples the input by a factor of 12, and the final outputs are temporally upsampled by a factor of 2 to align with the label frame rate of 10 fps. Given that the array response is almost frequency-independent up to about 9 kHz in the DCASE datasets, frequency bands above 9 kHz, corresponding to frequency bin indexes 384 and above, are linearly compressed by a factor of 8. As such, the frequency dimension of all linear-scale features is F=400. Unless stated otherwise, initial feature volumes are normalized globally to zero mean and unit standard deviation per channel. We use audio chunks of 8s in length for model training. The hyperparameter λ in Equation (Equation 6) for the attention module is set to $1\times 10^{-4}$, selected through hyperparameter search. A detailed analysis of λ is presented in the following subsection. We employ the Adam optimizer [44] for training. The learning rate is initially set at $3\times 10^{-4}$ and is linearly reduced to $1\times 10^{-5}$ over the final 35 epochs of the total 70 training epochs. A threshold of 0.4 is used to binarize active class predictions per track in the model outputs.

### 4.2. Experimental Results

#### 4.2.1. Choice of SELD Features

For a rational comparison of the performance of different input features, we implement the SELD network architecture described in [24] with its default settings. Note that we utilize the multi-ACCDOA format rather than the SELDnet format to output overlapping sound events from the same class. Table 3 shows the SELD performance of all considered features with data augmentation. The comprehensive metric $E_{\mathrm{SELD}}$ simplifies the evaluation of overall performance. The directional arrows in the table indicate whether an increase or decrease in a metric value signifies improvement.

As expected, features for FOA generally surpass those for MIC in most evaluation metrics. This suggests that the FOA format, with its built-in spatial audio encoding, is more appropriate for the SELD task. In detail, SALSA, SALSA-Lite, and IV-based features consistently perform better than GCC-based ones, underscoring the importance of maintaining TF alignment between feature maps. This alignment enables the network to effectively learn and discern multichannel event-level patterns in image-like input features. LinSpecIV exhibits superior performance compared to MelSpecIV, indicating that the use of a linear-scale may provide additional performance benefits. The performance of SALSA is relatively subpar for each format, most likely due to its focus on single-source TF bins, which results in the exclusion of important information from multisource TF bins. Among the features, SALSA-Lite achieves the lowest $E_{\mathrm{SELD}}$ value with micro- and macro-averages of 0.332 and 0.343 in the MIC format, as highlighted in Table 3. Likewise, LinSpecIV achieves the lowest $E_{\mathrm{SELD}}$ value with micro- and macro-averages of 0.331 and 0.335 in the FOA format. Both features take advantage of TF alignment and linear scale, making them ideal input features for subsequent experiments in their respective formats.

#### 4.2.2. Effect of Attention Modules

We evaluate SimAM on the SELD task against three representative attention modules (SE [14], CBAM [16], and GC [17]) and compare the performance of its enhanced CRNN with SOTA SELD systems using the L3DAS22 Task 2 dataset [38]. To adhere to the criteria of the L3DAS22 Challenge, we adapt the output format of CRNN to the track-wise format [11], which is necessary for estimating the exact locations of sound events. The performance is assessed using official evaluation metrics, and the results are shown in Table 4. The majority of the attention modules are capable of improving the performance of CRNN, in which SimAM stands out by achieving the highest F1-score without adding network parameters. In addition, SimAM-enhanced CRNN significantly outperforms the official baseline [38] across all evaluation metrics and even surpasses the second-ranked ensemble model [45] in the Challenge. Although it slightly underperformed compared to the top-ranked ensemble model [9], which benefited from training on both the L3DAS21 and L3DAS22 datasets with two FOA microphone arrays, the overall performance is still impressive. The results of these studies indicate that SimAM generates true 3D attention weights that effectively and efficiently enhance the representation of sound events in all three dimensions.

#### 4.2.3. Analysis of λ in the Attention Module

Figure 5 shows our search results for λ using CRNN on LinSpecIV features from the TNSSE 2021 development dataset. Given the impracticality of verifying all possible real values of λ, we restrict our search to a predetermined set ranging from $10^{-6}$ to $10^{-1}$, as shown on the *x*-axis of the figure. We repeat the process three times for each candidate λ value and report the means and standard deviations of the micro-averaged and macro-averaged $E_{\mathrm{SELD}}$ values. The results clearly indicate that SimAM significantly improves the performance of CRNN across the evaluated λ values, with $\lambda =10^{-4}$ emerging as the optimal value for superior SELD performance. Therefore, we select $\lambda =10^{-4}$ for implementation in our method.

#### 4.2.4. Experiments on a Synthetic Spatial Audio Dataset

Table 5 illustrates the performance of our newly developed STFF-Net and its control models for both MIC and FOA formats, as well as SOTA methods, on the test split of the TNSSE 2021 dataset [10]. The evaluations use both micro-average and macro-average metrics, the former for fair comparison with prior works and the latter for ease of comparison with future works. In terms of micro-average metrics, our STFF-Net outperforms the DCASE baseline SELDnet [10] by a substantial margin and surpasses several SOTA systems for the FOA format. In particular, STFF-Net exhibits a significant increase in $F_{\leq 20^{\circ }}$ and $LR_{\mathrm{CD}}$ by 23.8% and 18.1%, respectively, alongside a decrease in $ER_{\leq 20^{\circ }}$ by 23.5% and in $LE_{\mathrm{CD}}$ by 10.55 points compared to G-SELD [26]. Against HAAC-enhanced EINV2 [46], STFF-Net improves $F_{\leq 20^{\circ }}$ and $LR_{\mathrm{CD}}$ by 7.7% and 4%, respectively, and also reduces $ER_{\leq 20^{\circ }}$ by 7.5% and $LE_{\mathrm{CD}}$ by 3.31 points. Although marginally higher than the top-ranking ensemble model [7] from the DCASE 2021 Challenge in micro-averaged $E_{\mathrm{SELD}}$, STFF-Net achieves a higher $LR_{\mathrm{CD}}$ and lower $ER_{\leq 20^{\circ }}$ with fewer than half the parameters.

Our ablation studies, presented in the last two major rows of Table 5, quantify the effect of each module on SELD performance. In general, FOA models consistently outperform MIC models. The integration of SimAM significantly enhances the performance of CRNN over CBAM for both formats, and SimAM-enhanced CRNN achieves the best SELD results. This demonstrates that SimAM directly generates 3D attention weights to capture subtleties that may be neglected by CBAM’s two-step procedure, enabling more precise attention adjustment for target events without increasing network complexity. Furthermore, the SimAM-enhanced CRNN transforms into STFF-Net by substituting 2D convolutional modules with 3D ones. STFF-Net obtains the best macro-averaged $F_{\leq 20^{\circ }}$ and $LE_{\mathrm{CD}}$ among control models for the FOA format and the best $LR_{\mathrm{CD}}$ among control models for the MIC format, but the overall performance is slightly inferior to SimAM-enhanced CRNN. One potential explanation is that 3D convolutions incorporate an additional dimension to capture the evolving spatial distribution of sound events, thereby learning comprehensive features of sound events. However, the risk of overfitting is exacerbated by the complexity introduced by the additional dimension, particularly when the event categories are varied or the data are limited. Therefore, it is advisable to conduct additional assessments of STFF-Net with larger datasets.

Additionally, the corresponding class-wise performance analysis for the FOA format, shown in Figure 6, offers insight into the strengths and weaknesses of the systems with respect to each sound class. The $F_{\leq 20^{\circ }}$ and $LE_{\mathrm{CD}}$ metrics for half of the target classes, including Alarm, Crying Baby, and Female Speech, are significantly improved by the proposed STFF-Net compared to CRNN. These improvements collectively elevate the overall system performance. In contrast, the classes Footsteps and Knocking on Door present more of a challenge, as demonstrated by their lower $F_{\leq 20^{\circ }}$ and higher $LE_{\mathrm{CD}}$.

#### 4.2.5. Experiments on a Real Spatial Audio Dataset

Table 6 illustrates the performance of our STFF-Net and its control models for both MIC and FOA formats, as well as SOTA methods, on the test split of the STARSS 2022 dataset [37]. The models are trained using the training set of the STARSS 2022 dataset and augmented with synthesized simulation data from DCASE 2021 [10] to compensate for the limited duration of the real dataset. Despite the presence of challenging real-world conditions, STFF-Net still considerably outperforms the official baseline SELDnet across all metrics for both MIC and FOA formats. When compared to SOTA systems such as SwG-former [27], SwG-EINV2 [27], and HAAC-enhanced EINV2 [46], STFF-Net shows marked improvements in performance, as indicated by the lowest macro-averaged $E_{\mathrm{SELD}}$ and the best location-dependent SED metrics $ER_{\leq 20^{\circ }}$ and $F_{\leq 20^{\circ }}$, although $LE_{\mathrm{CD}}$ is slightly higher than HAAC-enhanced EINV2, and $LR_{\mathrm{CD}}$ is slightly lower than SwG-EINV2. In addition, STFF-Net is more compact in terms of model complexity than these competing models. Notably, STFF-Net for the FOA format improves macro-averaged $E_{\mathrm{SELD}}$ by 2.5% over the second-ranked ensemble model from the DCASE 2022 Challenge, while requiring fewer than one-third of the parameters. These results demonstrate that our proposed STFF-Net effectively learns comprehensive features of sound events to perform polyphonic SELD.

In the ablation study, there remains a large performance gap between the FOA and MIC models when applied in real-world audio contexts. SimAM enhances the performance of CRNN for both formats with a distinct advantage over CBAM. The SimAM-enhanced CRNN is further improved by incorporating 3D convolution modules instead of the original 2D ones, resulting in the STFF-Net. For the FOA format, STFF-Net performs better than the control models, as evidenced by a micro-averaged $E_{\mathrm{SELD}}$ of 0.309 and a macro-averaged $E_{\mathrm{SELD}}$ of 0.356. These improvements underscore the importance of comprehensive feature extraction of sound events in more varied and complex real-world environments. However, for the MIC format, STFF-Net shows a slight decline in performance. This decline is likely due to the necessity of a lower cutoff frequency in the MIC format to avoid spatial aliasing, which results in the loss of specific sound event information. Overall, our STFF-Net integrates all the proposed improvements.

Figure 7 shows the class-wise performance analysis of STFF-Net and control models for the FOA format on real spatial data. For most target classes, STFF-Net is superior to CRNN in both $F_{\leq 20^{\circ }}$ and $LE_{\mathrm{CD}}$ metrics, leading to better overall performance. Interestingly, the Footsteps and Knock classes show significant improvements, which differ from the findings obtained using the TNSEE 2021 dataset. However, the Faucet and Door classes exhibit a decline in performance, as evidenced by their respective $E_{\mathrm{SELD}}$ of 0.536 and 0.507, which are notably greater than the mean $E_{\mathrm{SELD}}$. One possible explanation for the Faucet class’s poor performance is the occurrence of false negatives caused by directional interference from other classes, such as Dishes, Pots, and Pans. Another consideration is that the Door class is rather sparse, comprising just 0.6% of all frames, which might affect the model’s ability to effectively learn this class.

To observe the transformation of features through the various stages of our STFF-Net, we utilize t-SNE [48] to visualize 3 s audio clips sampled every 3 min from the test set of the STARSS 2022 dataset. As shown in Figure 8, the scatter plot of the input features displays somewhat dispersed clustering of the same sound classes, suggesting that intrinsic patterns may exist within the spatio-temporal-frequency volume. As data flow through the backbone network, there is a discernible improvement in the segregation of different classes with reduced overlap, indicating that the backbone effectively extracts more pertinent features. Further processing through the RNN layers, which capture the temporal dependencies, results in sound classes becoming clearly distinguishable and tightly clustered. In the final FC output, the target classes are clearly separated with minimal overlap, reflecting the model’s ability to capture subtle differences between complex sound events. This separation underscores the effectiveness of the refined features, which are highly discriminative and well-suited for the SELD task.

## 5. Conclusions

In conclusion, this paper introduces a novel framework for polyphonic SELD that utilizes the newly developed Spatio-Temporal-Frequency Fusion Network (STFF-Net) to jointly learn comprehensive features of sound events from multichannel audio recordings. The Enhanced-3D (E3D) building block in our STFF-Net captures and refines intricate correlations across spatial, temporal, and frequency domains by combining 3D convolutions with non-parametric attention mechanisms. Additionally, we incorporate the multi-ACCDOA format to handle overlapping events from the same class. Our rigorous experimental evaluations on a range of benchmark datasets, both synthetic and real-world, demonstrate the effectiveness and generalization of the proposed STFF-Net. Significant enhancements in performance are readily apparent when compared to other SOTA methods. We hope this innovative approach to jointly learning comprehensive features of sound events will drive future advancements in acoustic scene analysis.

## Figures and Tables

**Figure 1 sensors-24-06090-f001:**
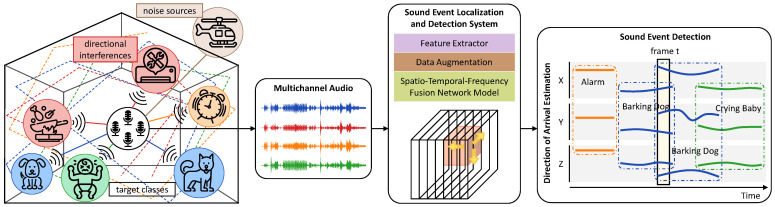
SELD task illustration. Our objective is to detect and localize the sources of interest given a set of target event classes and a multichannel audio signal recorded in conditions that include multiple concurrently active target sources, ambient noises, and directional interferences.

**Figure 2 sensors-24-06090-f002:**
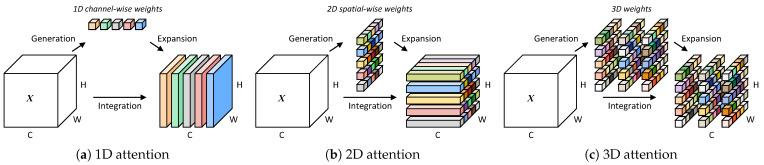
Comparative diagrams of different attention mechanisms. The 1D (**a**) and 2D (**b**) attention mechanisms generate weights that adjust channels or regions of the feature maps, respectively. In contrast, the 3D attention mechanism (**c**) generates weights capable of adjusting each individual point within the features. These weights are visually represented by colored bricks.

**Figure 3 sensors-24-06090-f003:**
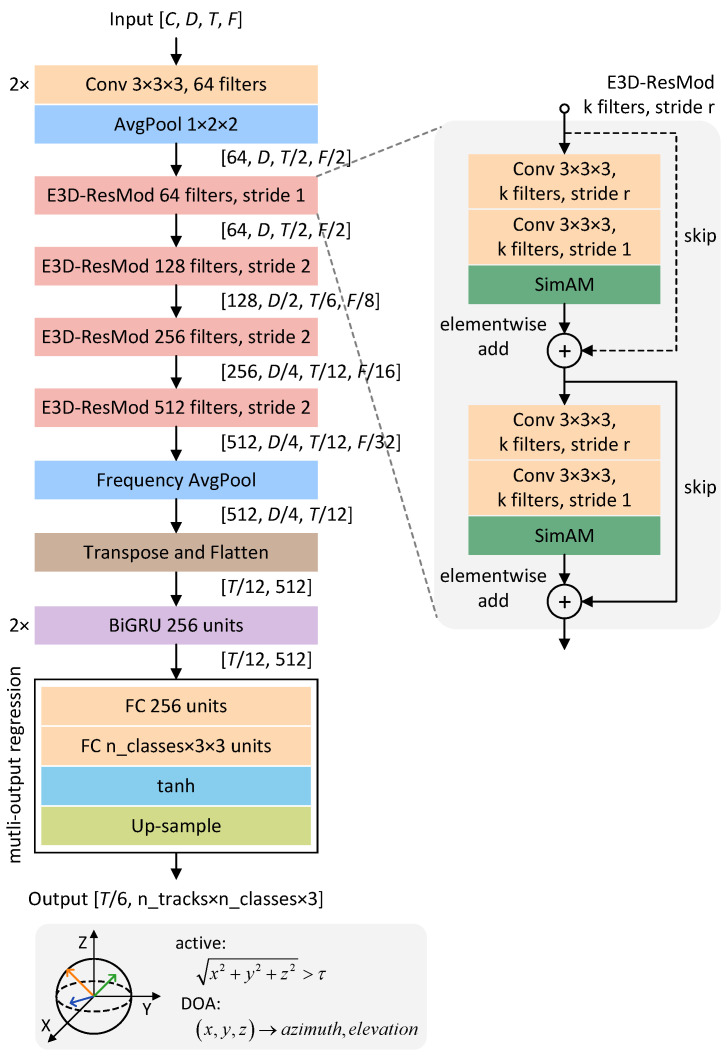
Architecture of our STFF-Net model for polyphonic SELD. The network adapts to different input feature volumes in FOA and MIC format.

**Figure 4 sensors-24-06090-f004:**
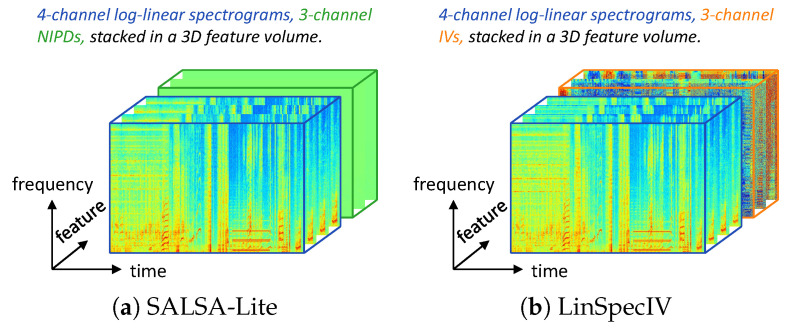
Spatio-temporal-frequency volumes for different audio formats. (**a**) Log-linear spectrograms and NIPDs for a four-channel MIC format audio clip, combined and aligned to create a 3D feature volume. (**b**) Log-linear spectrograms and IVs of an FOA format audio clip, combined and aligned to create a 3D feature volume.

**Figure 5 sensors-24-06090-f005:**
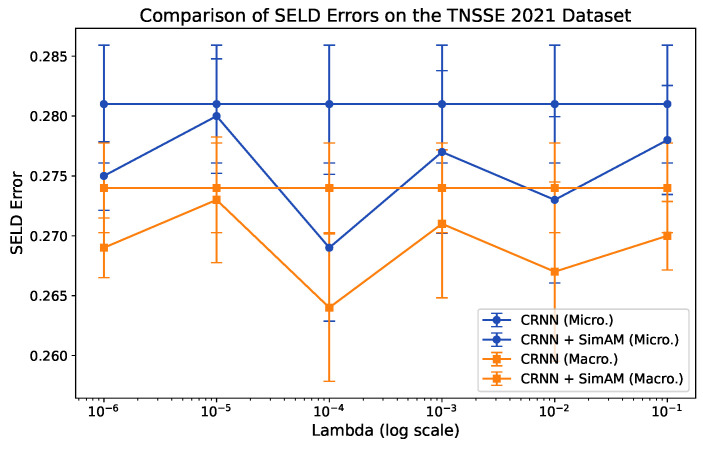
Visualization of the λ search for SELD performance. The analysis is conducted using CRNN on the testing portion of the development dataset. Means and standard deviations are calculated based on three separate trials.

**Figure 6 sensors-24-06090-f006:**
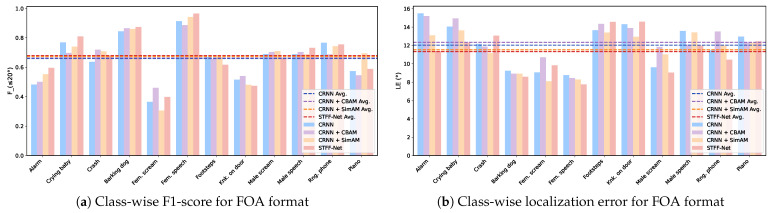
Class-wise performance comparisons for STFF-Net and control models on the test split of the TNSSE 2021 dataset. F1-score and localization error for both networks are presented on the left and right, respectively.

**Figure 7 sensors-24-06090-f007:**
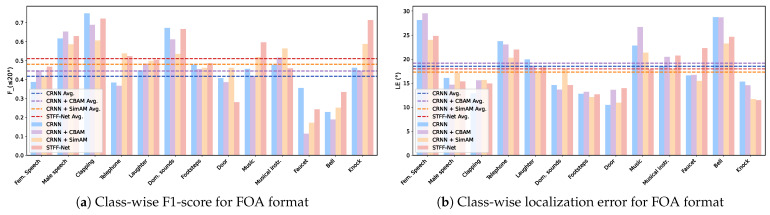
Class-wise performance comparisons for STFF-Net and control models on the test split of the STARSS 2022 dataset. F1-score and localization errors for both networks are presented on the left and right, respectively.

**Figure 8 sensors-24-06090-f008:**
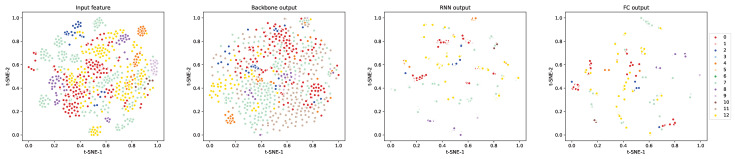
t-SNE visualization of input features, backbone outputs, RNN outputs, and FC outputs for selected test samples from the STARSS 2022 dataset. The colors denote different sound classes. Best viewed in color.

**Table 1 sensors-24-06090-t001:** Comparative analysis of different SELD features.

Feature Components	Format	Abbreviation	# Feature Channels	Scale	TF Alignment
Log-mel spectrogram, mel-scale GCC-PHAT	MIC	MelSpecGCC	4+6	mel	✗
Log-linear spectrogram, linear-scale GCC-PHAT	MIC	LinSpecGCC	4+6	linear	✗
Log-linear spectrogram, NIPD	MIC	SALSA-Lite	4+3	linear	✓
Log-linear spectrogram, normalized principal eigenvector	MIC/FOA	SALSA	4+3	linear	✓
Log-mel spectrogram, mel-scale IV	FOA	MelSpecIV	4+3	mel	✓
Log-linear spectrogram, linear-scale IV	FOA	LinSpecIV	4+3	linear	✓

The total number of feature channels is derived from four-channel audio recordings. GCC-PHAT stands for Generalized Cross-Correlation with Phase Transform.

**Table 2 sensors-24-06090-t002:** Characteristics of the TNSSE 2021, STARSS 2022, and L3DAS22 Task 2 datasets.

Characteristics	TNSSE	STARSS	L3DAS22
Spatial audio format	MIC, FOA	MIC, FOA	FOA
Sampling rate	24 kHz	24 kHz	32 kHz
Type of spatial recordings	synthetic	real	synthetic
Moving sources	✓	✓	✗
Ambient noise	✓	✓	✓
Reverberation	✓	✓	✓
Non-target interfering events	✓	✓	✗
Maximum degree of polyphony	3	5	3
Overlapping of same class events (%)	high	low	low
Number of target sound classes	12	13	14

**Table 3 sensors-24-06090-t003:** SELD performances of different features with data augmentation.

Feature	Cutoff Freq.	Micro-Average Metrics					Macro-Average Metrics				
		$↓{\mathrm{ER}}_{\leq 20^{\circ }}$	$↑F_{\leq 20^{\circ }}$	$↓{\mathrm{LE}}_{\mathrm{CD}}$	$↑{\mathrm{LR}}_{\mathrm{CD}}$	$↓E_{\mathrm{SELD}}$	$↓{\mathrm{ER}}_{\leq 20^{\circ }}$	$↑F_{\leq 20^{\circ }}$	$↓{\mathrm{LE}}_{\mathrm{CD}}$	$↑{\mathrm{LR}}_{\mathrm{CD}}$	$↓E_{\mathrm{SELD}}$
**MIC format**											
MelSpecGCC	12 kHz	0.616	0.466	22.2°	**0.720**	0.388	0.616	0.439	21.3°	0.685	0.402
LinSpecGCC	12 kHz	0.630	0.441	21.7°	0.681	0.407	0.630	0.411	20.9°	0.635	0.425
MIC SALSA	4 kHz	0.529	0.532	17.3°	0.690	0.351	0.529	0.473	18.5°	0.697	0.365
SALSA-Lite	2 kHz	**0.500**	**0.567**	**15.5**°	0.691	**0.332**	**0.500**	**0.515**	**16.6°**	**0.703**	**0.343**
**FOA format**											
MelSpecIV	12 kHz	0.498	0.564	15.8°	**0.685**	0.334	0.498	0.522	16.2°	**0.716**	0.338
LinSpecIV	12 kHz	**0.488**	**0.574**	**13.9°**	0.667	**0.331**	**0.488**	**0.525**	**15.2°**	0.708	**0.335**
FOA SALSA	9 kHz	0.503	0.560	15.4°	0.664	0.341	0.503	0.513	16.4°	0.695	0.347

**Table 4 sensors-24-06090-t004:** Performance comparison of baseline model and its enhancements with attention modules versus SOTA SELD methods on the test split of the L3DAS22 Task 2 dataset.

Approach	Params.	FOA Num.	Precision	Recall	F1-Score
(’22) revised SELDnet [38]	7.0 M	single	0.423	0.289	0.343
(’22 #1) Hu et al. [9] †	-	double	0.706	**0.691**	**0.699**
(’22 #2) Mao et al. [45] †	37.8 M	single	0.600	0.584	0.592
CRNN	13.6 M	single	0.700	0.586	0.638
CRNN + SE	13.7 M	single	**0.710**	0.588	0.643
CRNN + CBAM	13.7 M	single	0.705	0.585	0.640
CRNN + GC	13.7 M	single	0.702	0.584	0.638
CRNN + SimAM	13.6 M	single	0.701	0.595	0.644

The number following the ‘#’ symbol indicates the model’s rank in the challenge. The ‘†’ symbol denotes an ensemble model.

**Table 5 sensors-24-06090-t005:** Performance evaluation of SELD methods on the test split of the TNSSE 2021 dataset, including approach comparisons and module ablation studies.

Approach	Parameters	Format	Micro-Average Metrics					Macro-Average Metrics				
			$↓{\mathrm{ER}}_{\leq 20^{\circ }}$	$↑F_{\leq 20^{\circ }}$	$↓{\mathrm{LE}}_{\mathrm{CD}}$	$↑{\mathrm{LR}}_{\mathrm{CD}}$	$↓E_{\mathrm{SELD}}$	$↓{\mathrm{ER}}_{\leq 20^{\circ }}$	$↑F_{\leq 20^{\circ }}$	$↓{\mathrm{LE}}_{\mathrm{CD}}$	$↑{\mathrm{LR}}_{\mathrm{CD}}$	$↓E_{\mathrm{SELD}}$
(’21) SELDnet [10]	0.5 M	MIC	0.75	0.234	30.6°	0.378	0.577	0.75	-	-	-	-
		FOA	0.73	0.307	24.5°	0.405	0.539	0.73	-	-	-	-
(’21 #1) Shimada et al. [7] †	42 M	FOA	0.43	**0.699**	**11.1°**	0.732	0.265	0.43	-	-	-	-
G-SELD [26]	-	FOA	0.65	0.439	22.5°	0.559	0.444	0.65	-	-	-	-
HAAC-enhanced EINV2 [46]	85 M	FOA	0.49	0.60	15.26°	0.70	0.321	0.49	-	-	-	-
CRNN	13.7 M	MIC	0.453	0.624	14.24°	0.721	0.297	0.453	0.627	13.70°	0.738	0.291
CRNN + CBAM	13.8 M	MIC	0.476	0.605	15.04°	0.723	0.308	0.476	0.602	14.49°	0.737	0.304
CRNN + SimAM	13.7 M	MIC	**0.446**	**0.633**	**13.96°**	0.710	**0.295**	**0.446**	**0.633**	**13.11°**	0.731	**0.289**
STFF-Net	35.8 M	MIC	0.494	0.593	16.16°	**0.753**	0.310	0.494	0.609	15.39°	**0.755**	0.304
CRNN	13.7 M	FOA	0.411	0.660	12.67°	0.719	0.275	0.411	0.660	12.03°	0.740	0.270
CRNN + CBAM	13.8 M	FOA	0.407	0.663	12.93°	0.699	0.279	0.407	0.664	12.34°	0.735	0.269
CRNN + SimAM	13.7 M	FOA	**0.394**	0.682	12.12°	0.738	**0.260**	**0.394**	0.675	11.52°	**0.758**	**0.256**
STFF-Net	35.8 M	FOA	0.415	0.677	11.95°	**0.740**	0.266	0.415	**0.680**	**11.33°**	0.756	0.261

The number following the ‘#’ symbol indicates the model’s rank in the challenge. The ‘†’ symbol denotes an ensemble model.

**Table 6 sensors-24-06090-t006:** Performance evaluation of SELD methods on the test split of the STARSS 2022 dataset, including approach comparisons and module ablation studies.

Approach	Parameters	Format	Micro-Average Metrics					Macro-Average Metrics				
			$↓{\mathrm{ER}}_{\leq 20^{\circ }}$	$↑F_{\leq 20^{\circ }}$	$↓{\mathrm{LE}}_{\mathrm{CD}}$	$↑{\mathrm{LR}}_{\mathrm{CD}}$	$↓E_{\mathrm{SELD}}$	$↓{\mathrm{ER}}_{\leq 20^{\circ }}$	$↑F_{\leq 20^{\circ }}$	$↓{\mathrm{LE}}_{\mathrm{CD}}$	$↑{\mathrm{LR}}_{\mathrm{CD}}$	$↓E_{\mathrm{SELD}}$
(’22) SELDnet [37]	604 k	MIC	0.71	0.36	-	-	-	0.71	0.18	32.2°	0.47	0.560
		FOA	0.71	0.36	-	-	-	0.71	0.21	29.3°	0.46	0.551
(’22 #2) Hu et al. [47] †	85 M	FOA	0.53	-	-	-	-	0.53	0.481	17.8°	0.626	0.381
HAAC-enhanced EINV2 [46]	85 M	FOA	0.54	-	-	-	-	0.54	0.45	**17.20°**	0.62	0.391
SwG-former [27]	110.8 M	FOA	0.64	-	-	-	-	0.64	0.452	24.5°	0.657	0.416
SwG-EINV2 [27]	288.5 M	FOA	0.63	-	-	-	-	0.63	0.489	20.9°	**0.718**	0.385
CRNN	13.7 M	MIC	0.566	0.483	21.62°	0.756	0.362	0.566	0.417	21.38°	0.621	0.412
CRNN + CBAM	13.8 M	MIC	0.585	0.471	21.46°	0.744	0.372	0.585	0.413	20.52°	**0.632**	0.413
CRNN + SimAM	13.7 M	MIC	**0.545**	**0.515**	**19.27°**	0.737	**0.350**	**0.545**	**0.439**	**18.77°**	0.598	**0.403**
STFF-Net	35.8 M	MIC	0.613	0.447	23.31°	**0.793**	0.376	0.613	0.393	21.78°	0.631	0.428
CRNN	13.7 M	FOA	0.547	0.521	19.69°	0.792	0.336	0.547	0.417	18.50°	0.658	0.380
CRNN + CBAM	13.8 M	FOA	0.526	0.530	20.54°	0.763	0.337	0.526	0.445	19.14°	0.630	0.389
CRNN + SimAM	13.7 M	FOA	0.531	0.532	19.33°	0.771	0.334	0.531	0.481	17.31°	0.654	0.373
STFF-Net	35.8 M	FOA	**0.499**	**0.570**	**18.00°**	**0.794**	**0.309**	**0.499**	**0.511**	17.96°	0.663	**0.356**

The number following the ‘#’ symbol indicates the model’s rank in the challenge. The ‘†’ symbol denotes an ensemble model.

## Data Availability

Data are contained within the article.

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
