# Peer review of "Joint Spatio-Temporal-Frequency Representation Learning for Improved Sound Event Localization and Detection"

_sensors, 2024, doi:10.3390/s24186090_

Round 1

Reviewer 1 Report

Comments and Suggestions for Authors

The manuscript proposes a novel 3D multi-channel audio representation for sound event detection and localization. The authors effectively outline the research problem, methodology, and findings, providing a solid foundation for the study. However, several key areas require further clarification and development. Specific comments include:

1.       The author describes CRNN in Table 4, but there is insufficient information regarding the architecture and its significance.

2.       Tables 5 and 6 present comparable results between the proposed method (STFF-Net) and CRNN. However, STFF-Net has more than double the computational complexity for similar performance. The author needs to clarify the significance of the proposed method in this context.

3.       In Figure 8, the t-SNE plot displays highly scattered feature representations of the output. The author should provide an explanation of this result to enhance reader understanding.

Comments on the Quality of English Language

Minor corrections are needed. 

Reviewer 2 Report

Comments and Suggestions for Authors

In this manuscript, the authors present a Spatio-Temporal-Frequency Fusion network for sound event localization and detection task. In general, the manuscript is clearly structured and well written. I’m impressed by the solid and detailed work. There are several suggestions that can help authors further improve the quality of this manuscript.

1.        What is the concept of spatial domain? It is not clearly defined in traditional signal processing tasks. In my opinion, the authors may want to express the different acoustic information captured by multi-channel microphones in space. So, the authors are advised to provide a clear definition of this non-standardized concept in the manuscript for readers to understand.

2.        What does the mean of the output results [T/6,n_classes×3×3] about STFF-Net model? To be specific, what does 3*3 stand for? Why is temporal dimension reduced by 6 in output? The authors need to explain in detail the parameter of each dimension in output result. Meanwhile, the authors are suggested to provide the information of input and output shape in each layer.

3.     Why does the authors choice the non-parametric attention module rather than conventional strategy like SE, CBAM and GC? What are the unique advantages of non-parametric attention modules? The authors need to analyze through literature review in the ‘introduction’ part.

4.        Considering that the authors adopt joint learning protocol instead of the two-stage strategy for localization and detection task, the joint optimization function for classification and DOA should be provided. Meanwhile, the entire optimization procedure should also be presented through a flowchart or pseudo-code.

Round 2

Reviewer 1 Report

Comments and Suggestions for Authors

Thank you for addressing my comments and concerns. I have no further comments regarding the revised manuscript.

Comments on the Quality of English Language

Enough for publication.

Reviewer 2 Report

Comments and Suggestions for Authors

The manuscript is well improved. Authors responded to all the queries raised at the time of the previous round of review. In my view, the present revised manuscript can be accepted for publication.